# ROYAL SOCIETY
# OPEN SCIENCE

**Subject Category:**
Biology (whole organism)

behaviour/ecology

gelatinous zooplankton, hydrozoa, jellyfish, cichlid, Lake Tanganyika

**Author for correspondence:**
Kristina M. Sefc
e-mail: kristina.sefc@uni-graz.at

†Present address: Department of Collective Behavior, Max Planck Institute of Animal Behavior, Universitätsstraße 10, 78464 Konstanz, Germany

# Freshwater hydrozoan blooms alter activity and behaviour of territorial cichlids in Lake Tanganyika

Aneesh P. H. Bose†, Holger Zimmermann and Kristina M. Sefc

Institute of Biology, University of Graz, Universitätsplatz 2, 8010 Graz, Austria

 KMS, 0000-0001-8108-8339

Blooms of gelatinous zooplankton can represent dramatic environmental perturbations for aquatic ecosystems. Yet, we still know little about how blooms impact fitness-related behaviours of fish caught within their areas of effect, especially for freshwater systems. Here, we documented the behavioural impacts of freshwater hydrozoan (*Limnocnida tanganjicae*) blooms on a territorial cichlid (*Variabilichromis moorii*), as well as on the wider community of cichlids in a shallow-water rocky habitat of Lake Tanganyika. Compared with non-bloom conditions, *V. moorii* individuals in the midst of blooms reduced their swimming and territory defence activities (each by approx. 50%) but not their foraging or affiliative behaviours. Despite this reduction in activity, *V. moorii* could not entirely avoid being stung and preferred to remain closer to the rocky substrata as opposed to the more open demersal zone. Many other fishes similarly hid among the benthic substrata, changing the composition of the fish community in the demersal zone during bloom conditions. Reductions in activity could have multiple fitness-related implications for individual fish. Establishing the consequences of these behavioural changes is important for understanding the effects of gelatinous zooplankton blooms in freshwater systems.

## 1. Introduction

The factors that promote the rapid growth, i.e. 'blooms', of gelatinous zooplankton populations in our oceans and freshwater systems have been a focus of considerable research to date [1–5]. More recent research examining the interactions between fish populations and blooms of gelatinous organisms suggests that their relationships can be complex [6]. Blooms can be harmful to fish populations by injuring or directly killing fish

with their stinging cells, including their eggs and larvae, or by competing for the same food resources [7–10]. However, blooms may also be beneficial by providing an important food source for certain fishes [11–13], and even by providing nursery and foraging space for juveniles of other species [4,14,15]. Thus, the impact of gelatinous zooplankton blooms on individual fish and fish populations can be highly nuanced and depend on a wide range of ecological and species-specific factors.

To date, studies investigating the interactions between fish and gelatinous zooplankton have focused a great deal on predation and foraging. For example, mauve stinger jellyfish, *Pelagia noctiluca*, represent an important food source for the commercially important Mediterranean bogue, *Boops boops* [16]. Furthermore, substantial diet overlaps between certain fishes and jellyfish species have also been uncovered suggesting the potential for food competition [10]. Juvenile fish may also enjoy increased foraging opportunities by associating with gelatinous zooplankton, either by feeding on them directly or on the prey that they capture in their tentacles [4,14]. However, we currently have a poor understanding of how gelatinous zooplankton can impact other behaviours outside of a foraging context, such as social behaviour or parental care.

During our recent fieldwork in Lake Tanganyika, Zambia, we encountered several freshwater hydrozoan blooms of *Limnocnida tanganjicae* (henceforth 'jellyfish') and took advantage of these ephemeral occurrences to document the effects of high densities of these jellyfish on local shallow-water, territorial cichlid fishes. Currently, very little is known about freshwater jellyfish ecology or their impacts on freshwater fish species in general. The littoral zones of Lake Tanganyika are home to an extremely diverse community of demersal and benthic fishes displaying a wide variety of parental care systems, with many species caring extensively for non-pelagic offspring [17]. It was previously unknown how *L. tanganjicae* blooms affect fishes in Lake Tanganyika. On the one hand, high densities of *L. tanganjicae* medusae in the water column could physically impede the movement of fish and threaten them with their stinging cells, interfering with normal foraging, courtship and territory defence behaviours as the fish attempt to avoid being harmed. *L. tanganjicae* may also compete for the same zooplankton prey items that zooplanktivorous fishes rely on, or even opportunistically feed on juvenile fishes. On the other hand, *L. tanganjicae* blooms could represent an abundant, though ephemeral, food source for fishes. The *L. tanganjicae* blooms that entered our study quadrat, therefore, offered the rare opportunity to examine the consequences of jellyfish blooms on fish behaviour in an understudied system, namely freshwater lakes.

To this end, we recorded the behaviours of one focal fish species, *Variabilichromis moorii*, and did so during two very different environmental conditions: amidst *L. tanganjicae* blooms and also during normal, non-bloom conditions. We observed stark differences in the jellyfish abundance of the water column between days that we classified as 'non-bloom' (i.e. water column was virtually devoid of medusae) and as 'bloom' (i.e. water column was dense with medusae). *Variabilichromis moorii* is a socially monogamous and biparental species of cichlid fish, and like many Tanganyikan cichlids, *V. moorii* is highly territorial and ecologically constrained to rocky littoral habitats. Thus, it is unable to actively relocate to avoid jellyfish blooms, which can sometimes span areas covering several hundreds of square kilometres [18]; instead, male–female social pairs, as well as solitary territory-owners, remain on their territories during the blooms despite the large numbers of jellyfish that they encounter. *Variabilichromis moorii* pairs hold territories (1–4 m$^2$) containing several rocky crevices where they can take shelter, graze on algae and raise offspring. Here, we specifically investigated whether the fish exhibited any changes to their swimming activity, foraging, territory defence and within-pair social interactions when in the presence of high jellyfish densities. Next, we also investigated how jellyfish blooms affected the wider community of littoral fishes that lived in our study quadrat by monitoring the abundances of different fish species visible in the benthic and demersal zones during both bloom and non-bloom conditions.

# 2. Material and methods

## 2.1. Freshwater jellyfish of Lake Tanganyika

*Limnocnida tanganjicae* (Hydrozoa, Cnidaria) is the only documented species of jellyfish present in Lake Tanganyika [19] and is typically found throughout the oxygenated water column in the upper 100 m [18]. While not a 'true' scyphozoan jellyfish, *L. tanganjicae* is a major part of the zooplankton community in the lake [20] though it has received very little research attention relative to other fauna in Lake Tanganyika [21]. *Limnocnida tanganjicae* blooms are both temporally and spatially variable [21] and can be triggered

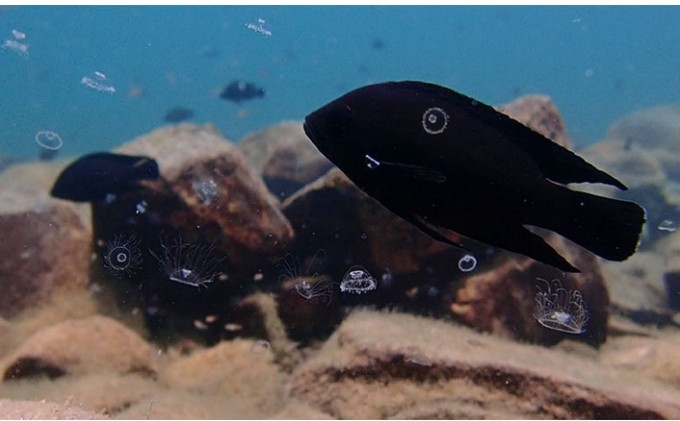

**Figure 1.** Field photograph of adult *Variabilichromis moorii* and fry on their territory amidst *Limnocnida tanganjicae* jellyfish. Photo credit: Aneesh P. H. Bose.

by internal seiching that leads to water mixing and upward nutrient fluxes [18,22]. A previous bloom was documented to have reached approximately 3000 individuals $m^{-3}$ by the coastline near Mpulungu, Zambia [18]. To avoid harmful UV radiation in the upper layers during day times with peak radiation, *L. tanganjicae* undergo low amplitude diel vertical migrations [18]. Exposure of *L. tanganjicae* to UV radiation at surface waters (top 5 m) during midday leads to mortality within 1 h [18].

## 2.2. Jellyfish bloom density and size distribution

Our study quadrat (approx. 1600 $m^2$, depth = 1.7–5.8 m) located by Mutondwe Island (8°42′29.4″ S 31°07′18.0″ E) contains dense breeding populations of a variety of cichlid species including *V. moorii*. On 17–18 September and then again on 26 September 2018, two *L. tanganjicae* blooms arrived at our study quadrat (figure 1). On 26 September, we estimated the density of the jellyfish bloom by performing four vertical hauls; we passed a 12 × 10 cm net (1600 µm mesh size) through 1.2 m of the water column starting at a depth of 5.5 m, which coincides with the depth of many *V. moorii* territories. We counted the number of jellyfish in the volume of water sieved by each haul and used these data to estimate the density of *L. tanganjicae* medusae $m^{-3}$ of the water column. One SCUBA diver photographed a haphazard set of 75 *L. tanganjicae* medusae drifting through the study quadrat. The photographs included a ruler for scale and we later used ImageJ (v. 1.50i) to measure the umbrella diameters of the individuals photographed.

## 2.3. Behavioural impacts of jellyfish blooms on *Variabilichromis moorii*

Between 17 September and 25 October 2018, we recorded 20 min videos of a total of 39 *V. moorii* territories while on SCUBA from a side-on perspective (Gopro Hero 5 session cameras, Olympus Tough TG5 digital cameras). We used these videos to score an array of behaviours and to quantify the average movement activity of each fish around their territories (details below). Most territories were held by a pair of *V. moorii* adults (*N* = 31 territories), though several were held by only a single female (*N* = 8 territories). We recorded the *V. moorii* territories according to a 2 × 2 factorial design. Approximately half of the territories were recorded in the midst of an *L. tanganjicae* bloom (*N* = 19 territories, *N* = 33 fish), while the other half were recorded under normal, non-bloom conditions (*N* = 20 territories, *N* = 37 fish). Because caring for offspring can have large effects on the movement and behavioural repertoire of parents, we crossed the jellyfish bloom conditions with the presence (*N* = 19 territories, *N* = 36 fish) or absence (*N* = 20 territories, *N* = 34 fish) of offspring on the territory. *Variabilichromis moorii* breed continually throughout the year and so any behavioural changes we detected here cannot be attributable to regular seasonal variation in behaviour.

We omitted the first 5 min of each 20 min video to avoid scoring behaviours and movement activity too soon after the deployment of the cameras. Using a custom program (written in Python by P. Nührenberg, Max Planck Institute for Animal Behaviour, Konstanz, Germany), we recorded the XY pixel coordinates of each territory-holding fish every 3 s throughout the remaining 15 min of each video (specifically, we extracted the coordinates of their visible eye in each frame). We also extracted each fish's total length in pixels and used this to calculate the number of body lengths that each fish

**Table 1.** Ethogram of behaviours scored from videos of territory-holding *Variabilichromis moorii*. Videos were recorded either amidst or outside of *Limnocnida tanganjicae* jellyfish bloom conditions and also with offspring being either present or absent from the territories.

| behaviour | description |
|---|---|
| feeding | Fish pecks at rock surface covered in algae. |
| nest defence | Fish lunges towards an unfamiliar con- or heterospecific intruding on their territory in order to drive them away. Defender may or may not make physical contact with the intruder. |
| social affiliation | One fish swims quickly towards its social partner and gently bumps them with their snout. The partner often responds by tilting their body laterally and sometimes quivers their body. |
| body scrape | Fish darts toward a rock surface and scrapes the side of its body along it. |
| jellyfish sting | Fish jolts violently and suddenly and often quivers its fins (this behaviour was only observed during jellyfish bloom conditions). |

swam in front of the camera for every 15 min video. Note that we subtracted any time that the fish was out of view of the camera, i.e. beyond edges of a picture frame or occluded by large rocky substrata, which was approximately $1.8 \pm 3.2$ min (mean ± s.d.) per video. From these data, we calculated each fish's average swimming speed in body lengths per second. Note that this method provides an estimate of movement in two dimensions only and we use this as a proxy for their overall swimming activity. Multiple behaviours are likely to contribute to the amount of swimming activity that each fish displays, including patrolling the territory, defending against intruders, foraging and interacting with their social partner. For each 15 min video, we also manually scored certain behaviours performed by the territory-holding *V. moorii* based on the ethogram presented in table 1.

## 2.4. Fish community during jellyfish bloom conditions

We surveyed the fish community on and near the substrata in the study quadrat during jellyfish bloom conditions (on 26 September 2018) and then again during normal, non-bloom conditions (on 27 and 28 September 2018) by swimming along 50 m transect lines. We set out two transects at each depth level of 5, 8 and 10 m and one transect at a depth of 3 m (for a total of seven transect lines in the quadrat) and swam these transect lines during both the bloom and non-bloom conditions (for a total of 14 transect swims). We waited 5 min after setting each 50 m line and then swam along it at a fixed elevation and recorded video of the swim facing forward (Olympus Tough TG5 digital camera). We swam at a constant pace and each transect took approximately 5 min to complete (mean ± s.d. = $290 \pm 36$ s). We recorded the number of fish visible in the transect videos by counting every fish that passed through an area estimated to be between 70 and 100 cm away from the lower edge of the camera's field of view. In this part of the frame, the width of the screen represents approximately 100 cm and thus every transect swim covers an area of approximately 50 m². We only counted fish heading towards the camera or entering the picture from the sides within the counting sector. We did not count any fish that were swimming away from the camera (i.e. in the direction of our swim) in order to minimize double counts. Only fish representing sub-adults or adults were counted and identified down to species level. Fish seen hiding under rocks and in crevices were not counted. *Telmatochromis vittatus* individuals were also omitted because they are small-bodied benthic fish that were typically partially occluded by rocky substrata in our videos and therefore could not be reliably counted. These surveys were meant to effectively capture changes in the visible abundances of non-pelagic species in the benthic and demersal zones between the two jellyfish bloom conditions. Non-pelagic species such as those that live in close proximity to the lake floor are unlikely to swim long distances to avoid jellyfish blooms and so any changes in their abundances that we record here are most likely due to these fish hiding among the substrata as opposed to fleeing the area.

## 2.5. Statistical analyses

All analyses were conducted in R [23]. We first investigated how *V. moorii* movement activity around their territories varied with high jellyfish densities and with offspring in their territory. We fit a linear

mixed-effects model (LMM; 'lme4' R package [24]) and included average fish swimming speed (continuous variable: body lengths per second) as the response variable. We included treatment group (four-level categorical variable: jellyfish presence/absence crossed with offspring presence/absence) as the predictor variable. Because each territory typically contained two fish, we included territory ID as a random intercept. We compared the groups with one another using pairwise Tukey contrasts ('multcomp' R package, [25]).

Next, we investigated whether any *V. moorii* behaviours differed during jellyfish bloom conditions and with the presence of offspring. We fit generalized linear mixed-effects models (GLMM), assuming a Poisson error distribution, and included the number of behaviours performed by each territory-holding fish as the response variable (count variables: nest defence, social affiliation, feeding and body scrapes). We also included treatment group (four-level categorical variable: jellyfish presence/absence crossed with offspring presence/absence) as the predictor variable. Again, we included a random intercept of territory ID and also controlled for overdispersion with an observation-level random intercept [26]. The total amount of time that each fish was visible to the camera (log-transformed) was included as a model offset. Again, we used pairwise Tukey contrasts to compare the groups with one another.

From each of our transect swims, we calculated total fish abundance (i.e. total number of fish seen) and species richness (i.e. number of species seen). We then tested whether these counts differed between the jellyfish bloom and non-bloom conditions. First, we fit a GLMM assuming a negative-binomial error distribution (to account for overdispersion) to the data on total fish abundance. Because we swam each transect twice, once during the jellyfish bloom and again during the non-bloom condition, we included 'transect ID' as a random intercept in this model. Then we fit a generalized linear model (GLM) assuming a Poisson error distribution to the data on species richness. Note that 'transect ID' did not account for any variance in this second model, which is why we dropped it as a random effect here. In both models, jellyfish condition (categorical variable: jellyfish bloom versus non-bloom) was included as a predictor variable.

Finally, we closely inspected the species-specific differences underlying the fish community shifts that we observed during our transect swims in the jellyfish bloom and non-bloom conditions. To do this, we first implemented a multivariate principal coordinate analysis (PCoA, also referred to as classic multi-dimensional scaling) on the observed fish community data (total fish counts per species summed across all transects per jellyfish condition) using a Bray–Curtis dissimilarity matrix ([27]; 'vegan' R package [28]). We tested for a statistical difference between the fish communities that we observed during our two sampling conditions with a permutation-based analysis of variance using 10 000 permutations of the data ('adonis2' function in 'vegan'). We then conducted a similarity percentages analysis (SIMPER) to identify which species were most influential in driving the community-level shifts. We followed this up by running individual Wilcoxon–Pratt signed-rank tests (suitable for paired data) for each recorded species.

## 3. Results

Average *L. tanganjicae* density measured on 26 September 2018 was $2448 \pm 519$ (mean ± s.e.) medusae $m^{-3}$. Average umbrella diameter of the jellyfish was 6.7 mm (range = 2.4–29.9 mm, median = 5.6 mm). We never observed *V. moorii* individuals consuming or attempting to consume *L. tanganjicae* despite many of the jellyfish being small enough to fit within the gapes of the fish. During jellyfish bloom conditions, the territory-holding *V. moorii* were also stung on average 3.4 times per 15 min video (range = 0–12, median = 2; electronic supplementary material, video).

Territory-holding *V. moorii* moved on average $0.32 \pm 0.18$ (mean ± s.d.) body lengths per second. Swimming activity was significantly reduced during jellyfish bloom conditions but did not vary with the presence of offspring on the territory (figure 2*a*; electronic supplementary material, table S1). The fish engaged in an average of 7.1 defence actions per 15 min video (range = 0–24, median = 6). The number of defence actions exhibited by the fish towards territory intruders was elevated when offspring were present, but this was only the case in non-bloom conditions (figure 2*b*; electronic supplementary material, table S1). Social partners engaged in an average of 1.5 social affiliative behaviours per 15 min video (range = 0–16, median = 1; N = 31 after omitting territories where only one adult was present). The number of social affiliative behaviours was highest when offspring were absent from the territory, and particularly when jellyfish were also absent (figure 2*c*; electronic supplementary material, table S1). *V. moorii* engaged in 0.6 body scrapes (range = 0–8, median = 0) and 9 feeding behaviours (range = 0–50, median = 4.5) on average per 15 min video, but the frequency of

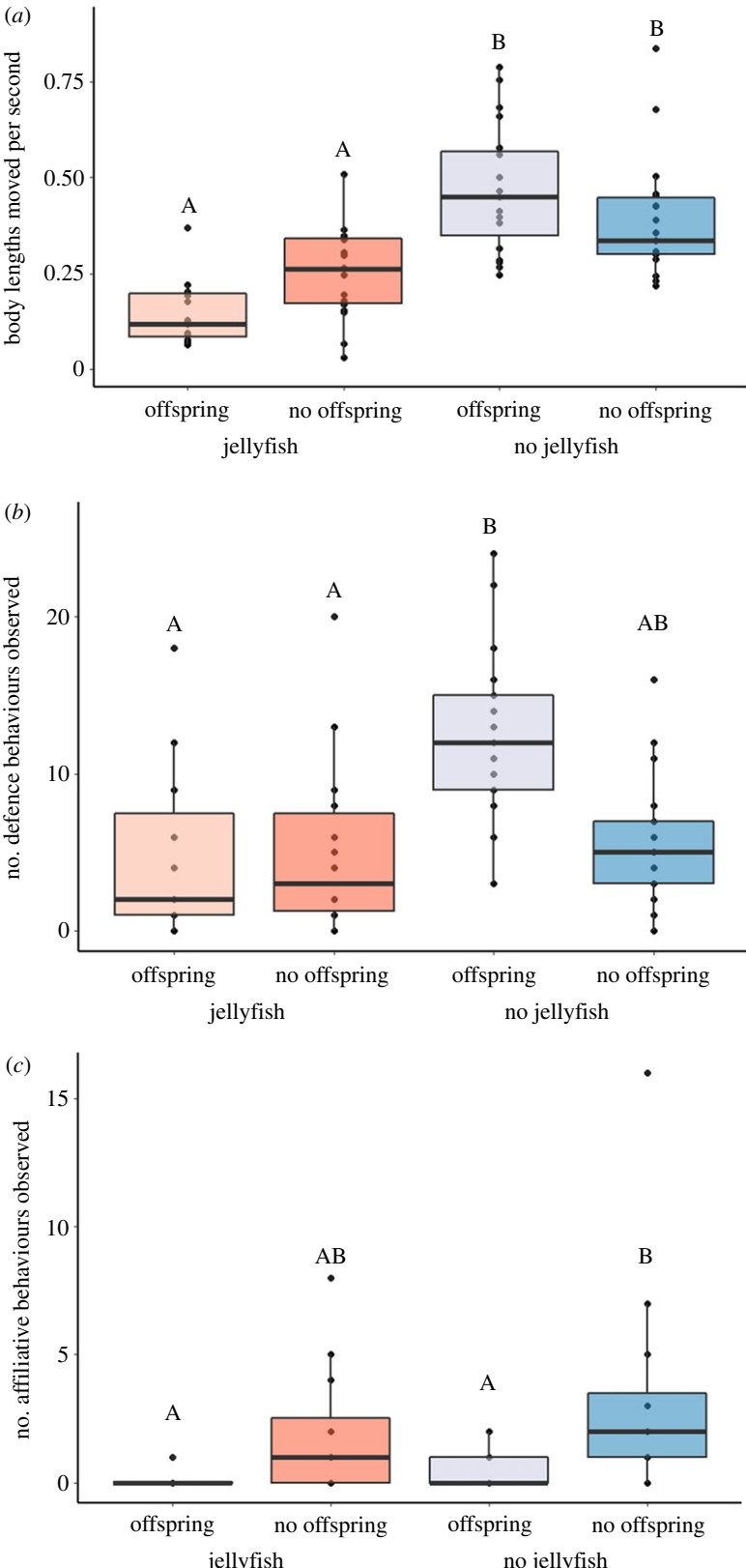

**Figure 2.** (*a*) Average movement activity of territory-holding *Variabilichromis moorii*. (*b*) Number of defence behaviours observed by each *V. moorii* towards foreign con- and heterospecifics intruding on their territories. (*c*) Number of affiliative behaviours observed between *V. moorii* social partners. Observations were taken over 15 min periods either amidst or outside of *Limnocnida tanganjicae* jellyfish bloom conditions and either with or without offspring on the territories. Letters indicate significant differences between groups (at $p < 0.05$) from linear mixed effects or generalized linear mixed-effects models. Note that the affiliative behaviours from (*c*) omit territories where only one adult was present ($N = 8$ out of 39 territories).

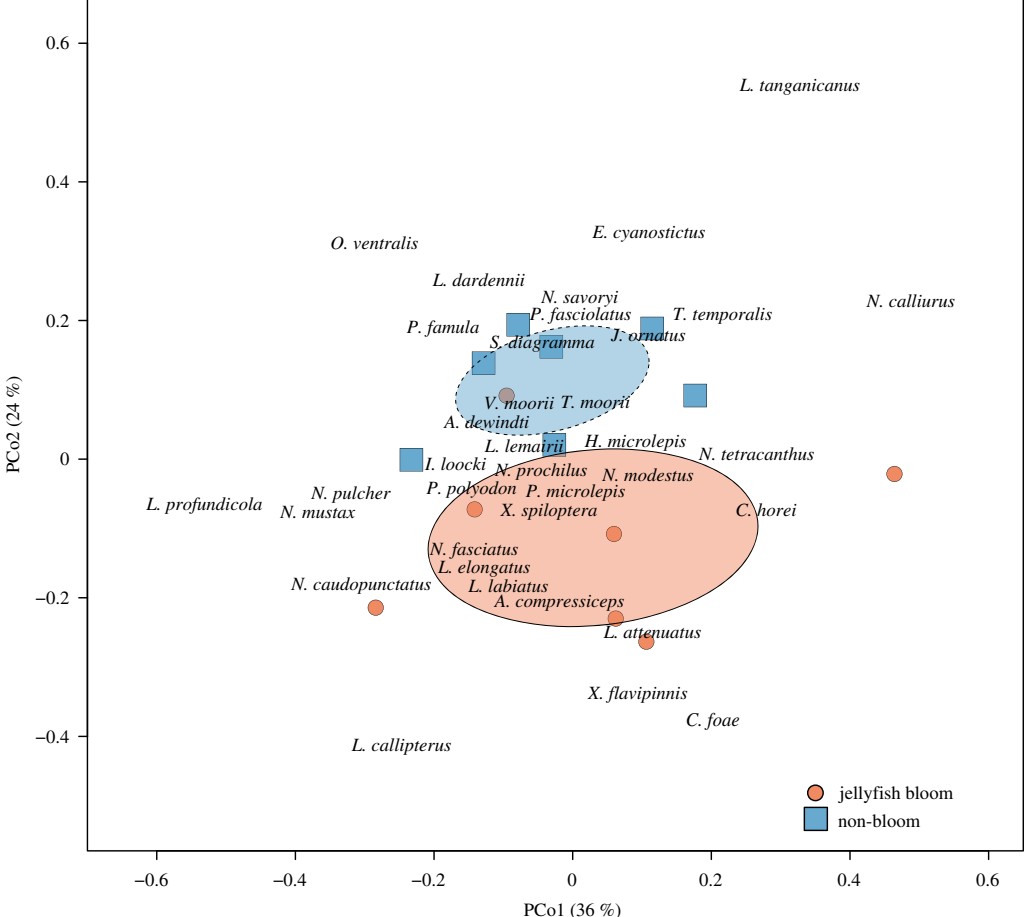

**Figure 3.** Principal coordinate analysis (PCoA from 'vegan' R package) biplot showing fish community differences between jellyfish bloom and non-bloom conditions within our study quadrat. Only the first two principal coordinate axes were considered significant as tested with a broken stick model. Ellipses illustrate one standard deviation of the point distributions for each group. Note that the species names have been jittered by a small margin to increase readability. PCo1 primarily describes variation across depth levels whereas PCo2 describes variation between jellyfish bloom conditions. Blue squares and red circles represent transects from which the fish community data were collected.

these behaviours did not vary with jellyfish bloom conditions or with the presence of offspring (electronic supplementary material, table S1).

We observed more fish in total (GLMER, Est. ± s.e. = 0.45 ± 0.07, $z = 6.19$, $p < 0.0001$) and a greater species richness (GLM, Est. ± s.e. = 0.26 ± 0.12, $z = 2.15$, $p = 0.032$) during the non-bloom conditions compared with the jellyfish bloom conditions. The overall fish community composition in the demersal zone differed between the bloom and non-bloom conditions (permutation test, $F_{1,12} = 2.52$, $p = 0.021$, figure 3) and had a dissimilarity measure of 45.4%. The five most influential species in driving the community-level differences were *Neolamprologus savoryi*, *Variabilichromis moorii*, *N. pulcher*, *N. caudopunctatus* and *N. tetracanthus*, respectively, contributing 17.9%, 14.6%, 11.4%, 9.0% and 7.8% to the overall dissimilarity score (table 2 for species counts and results of SIMPER analysis).

## 4. Discussion

In this study, we show that near-shore *L. tanganjicae* jellyfish blooms can alter the activity levels and behaviours of cichlid fishes caught within their areas of effect. Many of the fishes that live and breed in these shallow rocky habitats are spatially restricted to the few square metres of substrata comprising their territories (territories of most species span less than 5 m² [29], but sometimes reach up to 10 m in diameter [30]), and many of these fish remain on their territories as jellyfish blooms arrive. We recorded the density of one bloom to be approximately 2500 medusae m⁻³, similar to a bloom documented by Salonen *et al*. [18] that reached approximately 3000 medusae m⁻³. The jellyfish

**Table 2.** Fish counts for each species observed under jellyfish bloom and non-bloom conditions. Counts presented here are pooled across transects (seven 50 m transects per condition). Note that *Telmatochromis vittatus* individuals were not counted (see Material and methods). *p*-values indicate the results of Wilcoxon–Pratt signed-rank tests (for paired data) conducted for each species separately. Species-specific contributions to the total community dissimilarity score (based on SIMPER analysis) are presented in the right-most column.

| species | fish counts | | increase (+) or decrease (−) in abundance during bloom | *p*-value | dissimilarity contribution (%) |
| | non-bloom | jellyfish bloom | | | |
| --- | --- | --- | --- | --- | --- |
| *Altolamprologus compressiceps* | 5 | 4 | − | 0.56 | 0.69 |
| *Aulonocranus dewindti* | 17 | 5 | − | 0.06 | 1.52 |
| *Ctenochromis horei* | 5 | 7 | + | 0.79 | 0.87 |
| *Cyathopharynx foai* | 5 | 7 | + | 0.86 | 1.06 |
| *Eretmodus cyanostictus* | 22 | 5 | − | 0.06 | 2.17 |
| *Haplotaxodon microlepis* | 1 | 0 | − | 0.32 | 0.13 |
| *Interochromis loocki* | 44 | 18 | − | 0.06 | 3.93 |
| *Julidochromis ornatus* | 18 | 9 | − | 0.17 | 1.41 |
| *Lamprichthys tanganicanus* | 1 | 0 | − | 0.32 | 0.1 |
| *Lamprologus callipterus* | 3 | 7 | + | 0.16 | 0.86 |
| *Lamprologus lemairii* | 4 | 2 | − | 0.32 | 0.47 |
| *Lepidiolamprologus attenuatus* | 9 | 6 | − | 0.48 | 1.31 |
| *Lepidiolamprologus elongatus* | 16 | 12 | − | 0.44 | 1.69 |
| *Lepidiolamprologus profundicola* | 1 | 0 | − | 0.32 | 0.1 |
| *Limnotilapia dardennii* | 9 | 0 | − | 0.02 | 0.97 |
| *Lobochilotes labiatus* | 9 | 17 | + | 0.19 | 1.58 |
| *Neolamprologus calliurus* | 2 | 0 | − | 0.32 | 0.26 |
| *Neolamprologus caudopunctatus* | 85 | 107 | + | 0.55 | 9.03 |
| *Neolamprologus fasciatus* | 30 | 29 | − | 0.93 | 2.87 |
| *Neolamprologus modestus* | 27 | 19 | − | 0.39 | 1.87 |
| *Neolamprologus mustax* | 4 | 1 | − | 0.16 | 0.45 |
| *Neolamprologus prochilus* | 1 | 0 | − | 0.32 | 0.14 |
| *Neolamprologus pulcher* | 117 | 70 | − | 0.44 | 11.37 |
| *Neolamprologus savoryi* | 229 | 75 | − | 0.02 | 17.93 |
| *Neolamprologus tetracanthus* | 130 | 104 | − | 0.27 | 7.85 |
| *Ophthalmotilapia ventralis* | 11 | 0 | − | 0.05 | 1.12 |
| *Perissodus microlepis* | 15 | 13 | − | 0.80 | 1.68 |
| *Petrochromis famula* | 8 | 2 | − | 0.05 | 0.78 |
| *Petrochromis fasciolatus* | 4 | 4 | = | 0.48 | 0.72 |
| *Petrochromis polyodon* | 6 | 3 | − | 0.18 | 0.51 |
| *Simochromis diagramma* | 9 | 2 | − | 0.14 | 0.93 |
| *Telmatochromis temporalis* | 31 | 28 | − | 0.80 | 4.27 |
| *Tropheus moorii* | 17 | 10 | − | 0.39 | 1.29 |
| *Variabilichromis moorii* | 282 | 174 | − | 0.04 | 14.63 |
| *Xenotilapia flavipinnis* | 1 | 2 | + | 0.91 | 0.31 |
| *Xenotilapia spiloptera* | 48 | 45 | − | 0.80 | 3.13 |

blooms we observed were transient, lasting 1–2 days each, though long enough for us to observe behavioural effects among the fish in our study quadrat. A jellyfish bloom represents an ephemeral but intense environmental perturbation, and understanding how fish cope with or benefit from its effects is important in this understudied research system.

*Variabilichromis moorii* responded to *L. tanganjicae* blooms primarily by reducing their overall activity levels on their territories, presumably to avoid coming in contact with the jellyfish's tentacles. Adult fish were sometimes observed making sudden jolting or quivering responses during the jellyfish bloom conditions that we did not observe during the non-bloom conditions, suggesting that *V. moorii* can perceive the stings of *L. tanganjicae* (electronic supplementary material, video). Jellyfish stings have been linked to mass mortality events of farmed fish following gill damage, bacterial infections of skin lesions and metabolic disturbances [31–33]. Owing to the potential skin damage caused by jellyfish stings, we expected *V. moorii* adults to also increase their rate of body scraping, as scraping is a common behavioural response to skin irritation in fishes [34]. However, body scraping was performed rather infrequently (median body scrapes per 15 min video = 0) and we did not observe a clear relationship between body scraping and jellyfish presence. Anecdotally, we noted adult fish bearing white markings on their faces and flanks during the jellyfish blooms probably indicating the locations where they had been stung.

We did not detect any statistically clear effects of the jellyfish bloom conditions on either foraging or social affiliative behaviours in *V. moorii*. Feeding may not be compromised in *V. moorii* since this species predominantly grazes on algae that grow abundantly on the rock surfaces in its territories. Interestingly, *L. tanganjicae* were never consumed by *V. moorii* despite many medusae being small enough to fit within the gapes of adult fish. This suggests that *L. tanganjicae* blooms may not confer benefits through increased foraging opportunities to *V. moorii* or similar species. Social affiliation between the male–female social partners did not change clearly with the jellyfish conditions but was elevated when there were no offspring present on their territories. Affiliative behaviours are thought to strengthen and maintain social bonds between paired mates, family members or group members and can also buffer against stress in many mammal species [35,36]. More recently, social cues from conspecifics have been shown to buffer against stress in fish as well [37]. We originally predicted affiliative behaviours to show either of two opposing patterns: either an increase in frequency during jellyfish blooms to help counteract the stressful conditions, or a decrease in frequency mirroring the general reduction in activity levels. Though neither of these cases were supported, our results do suggest that affiliative behaviours are important for *V. moorii* pairs that are in between broods and not actively providing parental care.

*Variabilichromis moorii* engaged in approximately 48% fewer defensive behaviours during the jellyfish blooms than during the non-bloom conditions, and this difference was most notable when parents had offspring present on their territories. *V. moorii* pairs must defend against myriad of con- and heterospecifics that intrude on their territories, many of which prey upon offspring or compete for territory space [38]. Under non-bloom conditions, nearly all intrusions of the territory elicit a defensive reaction from one of the territory-holders to drive the foreign fish away [39]. Thus, the reduction in defensive behaviours observed here is most likely due to a decrease in intrusion pressure from the surrounding fishes as opposed to a lowered motivation to defend by the territory-holders. Either way, such a reduction in defensive behaviours may also explain some of the reduction in swimming activity that we describe above. While blooms may ease the intrusion pressure on brood-tending parents, the jellyfish themselves may form a new predation threat for young fish that could need guarding against. We observed one occurrence of a jellyfish that had drifted among a group of fry being taken up into the mouth of a parent *V. moorii* and then being spit out on the periphery of the territory. We also observed several offspring get captured in the tentacles of larger *L. tanganjicae* (A.P.H.B., H.Z. 2018, personal observations), though we were not able to take quantitative measurements of fry losses.

The idea that territories receive fewer intrusions by foreign fish during jellyfish blooms is consistent with our transect surveys, which suggest that many species of fish in the wider community are less active—and less visible—during bloom conditions. Numerous species declined in apparent abundance during the jellyfish bloom conditions. Rather than populating the water column, many fish during our transect surveys were seen hiding within crevices in the rocky substrata. In addition, our transect surveys captured additional species responses beyond variation in abundance. In particular, we documented similar numbers of *Xenotilapia spiloptera* in both jellyfish conditions (table 2); however, the *X. spiloptera* individuals visible during the blooms were often seen sitting on the substrata exhibiting dull and mottled body coloration indicative of stress.

Reductions in activity levels could have multiple fitness-related implications for individuals, including reduced opportunities for dispersal, mate search, courtship and feeding (in the case of open-water foragers), while at the same time representing a temporary respite from the energetic costs of patrolling and defence for territorial and brood-tending fishes. Apparent abundances of the most common territorial planktivores *N. pulcher* and *N. savoryi* [40] declined markedly during jellyfish blooms (table 2). This suggests that planktivores may especially reduce their activity during jellyfish blooms, possibly because they suffer from trophic competition with jellyfish [18] and also risk jellyfish stings during foraging. By contrast, the abundances of other open-water foragers such as the scale-eating *P. microlepis* and the piscivorous *Lepidiolamprologus* species [41] did not seem to be affected by the jellyfish blooms (table 2, though their counts were far lower overall).

Tanganyikan cichlids exhibit a wide variety of courtship and spawning behaviours, some of which require mobility that could be impaired during jellyfish blooms. For instance, polygynous mating in *O. ventralis*, *A. dewindtii* and *C. foai* (table 2) involves females that visit multiple males, who then approach and court them in the water column [41,42]. Jellyfish blooms may also affect cuckoldry, which is a common component of many Tanganyikan cichlid mating systems (e.g. [17,43,44]). In general, cuckolder males seek to intrude on mating events and steal fertilizations from spawning males, but if their movement is restricted, they could suffer in terms of missed reproductive opportunities. As such, jellyfish blooms may reduce overall mating activity but also afford benefits to spawning males by alleviating pressure from sperm competitors.

Here, we show and discuss the effects of jellyfish blooms on littoral cichlid fishes in a tropical freshwater lake. The ecology of freshwater jellyfish, as well as their interactions with fish, is a severely understudied topic of research. Long-term field studies are now required to quantify the reproductive and fitness consequences of jellyfish blooms on breeding fish. It will be important for these studies to account for direct offspring losses due to jellyfish predation, the energetic costs and benefits of altered activity levels, changes to feeding and reproductive strategies and the potential immune consequences of jellyfish stings.

Ethics. All fieldwork was carried out with the ethical approval of the ethics committee of the University of Graz (permit number 39/50/63 ex 2018/19). Fieldwork was carried out with the permission of the Fisheries Department of Zambia and under study permits issued by the government of Zambia (SP 007216, SP 008735).

Data accessibility. Data available from the Dryad Digital Repository: https://dx.doi.org/10.5061/dryad.k6ck467 [45].

Authors' contributions. A.P.H.B. conceived and designed the study. A.P.H.B. and H.Z. conducted the fieldwork. H.Z. identified species from the transect videos. A.P.H.B. conducted the statistical analyses and wrote the manuscript with input from H.Z. and K.M.S. All authors gave final approval for publication.

Competing interests. Prof. Kristina Sefc is a Board Member of Royal Society Open Science.

Funding. This work was supported by the Austrian Science Fund (FWF, grant no. P 27605-B25 to K.M.S.).

Acknowledgements. We would like to thank the Zambian Department of Fisheries in Mpulungu, especially L. Makasa and T. Banda, as well as C. Katongo of the University of Lusaka for kindly supporting our research at Lake Tanganyika. We are also grateful to B. Mbao our boat driver. Thank you to the groups of M. Taborsky, T. Takahashi and A. Jordan for sharing their equipment and expertise, especially to D. Josi, J. Frommen, J. Flury, F. Heussler and H. Tanaka. We also thank P. Nührenberg for his custom-made Python program.

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
