## [Reviewer comments · Royal Society Open Science]

Review History

RSOS-191053.R0 (Original submission)

Review form: Reviewer 1 (José Riascos)

Is the manuscript scientifically sound in its present form?

Yes

Are the interpretations and conclusions justified by the results?

Yes

Is the language acceptable?

Yes

Do you have any ethical concerns with this paper?

No

Have you any concerns about statistical analyses in this paper?

No

Recommendation?

Accept with minor revision (please list in comments)

Comments to the Author(s)

This is a good, well written paper dealing with the behavioural impacts of a freshwater hydrozoan on a territorial cichlid and on the composition of cichlids assemblages. The main issue with the manuscript is that authors have chosen a highly debatable or wrong context to justify their study, which in my view, they don't need: the study is interesting and valid without establishing a parallel with fish-jellyfish interactions in coastal marine systems. Despite the diversity of studies suggesting that anthropogenic stressors facilitate jellyfish blooms and a global trend of increasing jellyfish blooms, two recent studies (Pitt et al. 2018; Sanz-Martín et al. 2016) have clearly shown that the evidence supporting those claims is weak and mainly based in a couple of cosmopolitan species. These species do not represent the huge diversity life forms included in the term "jellyfish", which depending on the authors may comprise between two (Cnidarians, Ctenophores) and five phyla (Cnidarians, Ctenophores, Molluscs, Polychaetes and Chordata-Salps and Larvaceans). Therefore, I strongly recommend not to use this parallel to justify the study or to discuss the results.

Specific comments

Title: It is too general; it does not reflect the scope of the study. The manuscript deals with the effect of a freshwater hydrozoan on the behaviour of cichlids in the Tanganyika Lake.

Introduction:

- I suggest either avoid the use of the term "jellyfish" or clearly define it in the context of this study

- I suggest, if available, to include information on the abundance seasonality of the *L. taganjicae* L34-63. See general comments and reconsider the need of this section.

L88. Please define -here or in material and methods- how a bloom and non-bloom condition was defined. It was not clear for me if "normal" or "non-bloom" conditions mean no medusas at all or a low density.

Results:

L243-244. This is an interesting result, suggesting that the medusae population was mainly comprised of juveniles -this, by the way was not discussed in the next section. I am wondering if there is differences between juvenile and adult medusa in prey patterns and interspecific competitive interactions.

Discussion:

Generally I suggest avoiding stating again the results in order to discuss about them. The discussion is a bit too long.

L 274-280: Please see my general comments.

L280. In my view this section should start here

L 307-310: This paragraph should be at the end. Of this section

L373-380: This sections sounds to me as "concluding remarks", please consider

L381-384: see general comments

Figures:

Figure 2. Please revise the term "affiative" in the X-label

References:

Pitt, K. A., Lucas, C. H., Condon, R. H., Duarte, C. M., & Stewart-Koster, B. (2018). Claims that anthropogenic stressors facilitate jellyfish blooms have been amplified beyond the available evidence: a systematic review. *Frontiers in Marine Science*, 5, 451.

Sanz-Martín, M., Pitt, K. A., Condon, R. H., Lucas, C. H., Novaes de Santana, C., & Duarte, C. M. (2016). Flawed citation practices facilitate the unsubstantiated perception of a global trend toward increased jellyfish blooms. *Global Ecology and Biogeography*, 25(9), 1039-1049.

Review form: Reviewer 2

Is the manuscript scientifically sound in its present form?

No

Are the interpretations and conclusions justified by the results?

Yes

Is the language acceptable?

Yes

Do you have any ethical concerns with this paper?

Yes

Have you any concerns about statistical analyses in this paper?

No

Recommendation?

Major revision is needed (please make suggestions in comments)

Comments to the Author(s)

This is a fascinating study on the impact of a poorly-known jellyfish species on fish behavior. Overall the manuscript is well-written, but reflects several outdated views on jellyfish blooms which, even if accurate, are not comparable to this system. Unlike highly impacted coastal marine areas, the author's study area is an ancient freshwater lake in which the jelly species in question is a native member of the ecosystem. While it is true that jellyfish blooms impact fish behavior in marine systems, framing this study in the context of 'rising jellyfish numbers' and the 'global rise of jellyfish' (claims that are now being questioned), detracts from the fascinating work being done here on an remarkable and unique system. Because the human-associated causes and consequences of marine blooms likely do not impact this system, and because the field of marine blooms is itself rapidly changing, the authors should re-frame their introduction and discussion to highlight their system's fascinating qualities. The fact that so little is known about freshwater jellyfish ecology, and this species in particular, should be celebrated. Further, I see no reason their findings are not applicable to marine systems, but the language and literature currently cited reflect a "jellyfish as negative" perspective when really these are all native species interacting in a fascinating way.

15: Jellyfish are also beneficial to many fish species, so the relationship is complicated and should not be reduced to one of purely threat, but of impact more generally: Unravelling the macro-evolutionary ecology of fish-jellyfish associations. Caution is especially advised for poorly-understood freshwater jellyfish like *Limnocyclus tatei*. Resource: life in the 'gingerbread house' Donal C. Griffin, Chris Harrod, Jonathan D. R. Houghton and Isabella Capellini
Published: 20 March 2019 <https://doi.org/10.1098/rspb.2018.2325>

23: What is your evidence that *Limnocyclus tatei* have a noticeable sting to fish?

36-46: Consider the following sources and re-frame more accurately:

Flawed citation practices facilitate the unsubstantiated perception of a global trend toward increased jellyfish blooms Marina Sanz-Martín Kylie A. Pitt Robert H. Condon Cathy H. Lucas Charles Novaes de Santana Carlos M. Duarte First published: 22 June 2016
<https://doi.org/10.1111/geb.12474>

Questioning the Rise of Gelatinous Zooplankton in the World's Oceans Robert H. Condon, William M. Graham, Carlos M. Duarte, Kylie A. Pitt, Cathy H. Lucas, Steven H.D. Haddock, Kelly R. Sutherland, Kelly L. Robinson, Michael N Dawson, Mary Beth Decker, Claudia E. Mills, Jennifer E. Purcell, Alenka Malej, Hermes Mianzan, Shin-ichi Uye, Stefan Gelcich and Laurence P. Madin *BioScience* Vol. 62, No. 2 (February 2012), pp. 160-169 (10 pages)

64: There are thousands of Hydrozoan in the marine environment, many more than there are scyphozoans.

72: Comparing scyphozoans to freshwater hydrozoans is not really accurate, especially when hydrozoans are so abundant in marine systems. A more accurate comparison would be across hydrozoans.

74: The vast majority of hydrozoan medusae do not sting people and likely do not sting fish. What evidence do you have that fish can even perceive the sting of this poorly-know species?

77: But they may also be eaten. See: A Paradigm Shift in the Trophic Importance of Jellyfish? Author links open overlay panel Graeme C. Hays¹ Thomas K. Doyle^{2,3} Jonathan D. R. Houghton⁴ Show more <https://doi.org/10.1016/j.jtree.2018.09.001>

78: You can't make generalizations like this about jellyfish. As a clade Medusozoa has had more time to evolve and diversify than all of vertebrates combined.

83: The word invaded is inappropriate in this context. They are native and this is part of their natural cycle.

242: Wow, that's a lot of jellies!

247: Was the "sting" inferred from a behavioral response?

290: Stressor--a more general word would be perturbation.

380-382: Jellyfish may also serve as nurseries for young fish species: Ecological and Societal Benefits of Jellyfish Authors and affiliations Thomas K. Doyle Graeme C. Hays Chris Harrod Jonathan D. R. Houghton Email author 1. 2. 3. 4. 5. 6. Chapter First Online: 28 September 2013

Citation 3 is for ctenophores, which are not jellyfish and is not related to the work being done here.

Decision letter (RSOS-191053.R0)

22-Aug-2019

Dear Dr Sefc,

The editors assigned to your paper ("Jellyfish blooms alter activity and behaviour of territorial fish") have now received comments from reviewers. We would like you to revise your paper in accordance with the referee and Associate Editor suggestions which can be found below (not

including confidential reports to the Editor). Please note this decision does not guarantee eventual acceptance.

Please submit a copy of your revised paper before 14-Sep-2019. Please note that the revision deadline will expire at 00.00am on this date. If we do not hear from you within this time then it will be assumed that the paper has been withdrawn. In exceptional circumstances, extensions may be possible if agreed with the Editorial Office in advance. We do not allow multiple rounds of revision so we urge you to make every effort to fully address all of the comments at this stage. If deemed necessary by the Editors, your manuscript will be sent back to one or more of the original reviewers for assessment. If the original reviewers are not available, we may invite new reviewers.

- Data accessibility

<http://datadryad.org/submit?journalID=RSOS&manu=RSOS-191053>

- Competing interests

- Authors' contributions

- Acknowledgements

- Funding statement

on behalf of Dr Punidan Jeyasingh (Associate Editor) and Kevin Padian (Subject Editor)
openscience@royalsociety.org

Associate Editor's comments (Dr Punidan Jeyasingh):

This paper reports very interesting observations about the interactions between freshwater jellyfish and the behavior of fish in the African Great Lakes. The manuscript was assessed by two expert reviewers. Both reviewers were excited about the study. However, both felt (among other things) that the authors relied too much on the marine literature. I agree the elegance and importance of this nascent study, is its uniqueness in terms of study system and behavioral/ecological interactions. I felt the reviews were clear, fair, and constructive. With much gratitude to the expert reviewers, I invite the authors to address these comments.

Reviewers' Comments to Author:

Reviewer: 1

This is a good, well written paper dealing with the behavioural impacts of a freshwater hydrozoan on a territorial cichlid and on the composition of cichlids assemblages. The main issue

with the manuscript is that authors have chosen a highly debatable or wrong context to justify their study, which in my view, they don't need: the study is interesting and valid without establishing a parallel with fish-jellyfish interactions in coastal marine systems. Despite the diversity of studies suggesting that anthropogenic stressors facilitate jellyfish blooms and a global trend of increasing jellyfish blooms, two recent studies (Pitt et al. 2018; Sanz-Martín et al. 2016) have clearly shown that the evidence supporting those claims is weak and mainly based in a couple of cosmopolitan species. These species do not represent the huge diversity life forms included in the term "jellyfish", which depending on the authors may comprise between two (Cnidarians, Ctenophores) and five phyla (Cnidarians, Ctenophores, Molluscs, Polychaetes and Chordata-Salps and Larvaceans). Therefore, I strongly recommend not to use this parallel to justify the study or to discuss the results.

Specific comments

Title: It is too general; it does not reflect the scope of the study. The manuscript deals with the effect of a freshwater hydrozoan on the behaviour of cichlids in the Tanganyika Lake.

Introduction:

- I suggest either avoid the use of the term "jellyfish" or clearly define it in the context of this study

- I suggest, if available, to include information on the abundance seasonality of the *L. taganjicae* L34-63. See general comments and reconsider the need of this section.

L88. Please define -here or in material and methods- how a bloom and non-bloom condition was defined. It was not clear for me if "normal" or "non-bloom" conditions mean no medusas at all or a low density.

Results:

L243-244. This is an interesting result, suggesting that the medusae population was mainly comprised of juveniles -this, by the way was not discussed in the next section. I am wondering if there is differences between juvenile and adult medusa in prey patterns and interspecific competitive interactions.

Discussion:

Generally I suggest avoiding stating again the results in order to discuss about them. The discussion is a bit too long.

L 274-280: Please see my general comments.

L280. In my view this section should start here

L 307-310: This paragraph should be at the end. Of this section

L373-380: This sections sounds to me as "concluding remarks", please consider

L381-384: see general comments

Figures:

Figure 2. Please revise the term "affiative" in the X-label

References:

Pitt, K. A., Lucas, C. H., Condon, R. H., Duarte, C. M., & Stewart-Koster, B. (2018). Claims that anthropogenic stressors facilitate jellyfish blooms have been amplified beyond the available evidence: a systematic review. *Frontiers in Marine Science*, 5, 451.

Sanz-Martín, M., Pitt, K. A., Condon, R. H., Lucas, C. H., Novaes de Santana, C., & Duarte, C. M. (2016). Flawed citation practices facilitate the unsubstantiated perception of a global trend toward increased jellyfish blooms. *Global Ecology and Biogeography*, 25(9), 1039-1049.

Reviewer: 2

Comments to the Author(s)

This is a fascinating study on the impact of a poorly-known jellyfish species on fish behavior. Overall the manuscript is well-written, but reflects several outdated views on jellyfish blooms which, even if accurate, are not comparable to this system. Unlike highly impacted coastal marine areas, the author's study area is an ancient freshwater lake in which the jelly species in question is

a native member of the ecosystem. While it is true that jellyfish blooms impact fish behavior in marine systems, framing this study in the context of 'rising jellyfish numbers' and the 'global rise of jellyfish' (claims that are now being questioned), detracts from the fascinating work being done here on an remarkable and unique system. Because the human-associated causes and consequences of marine blooms likely do not impact this system, and because the field of marine blooms is itself rapidly changing, the authors should re-frame their introduction and discussion to highlight their system's fascinating qualities. The fact that so little is known about freshwater jellyfish ecology, and this species in particular, should be celebrated. Further, I see no reason their findings are not applicable to marine systems, but the language and literature currently cited reflect a "jellyfish as negative" perspective when really these are all native species interacting in a fascinating way.

15: Jellyfish are also beneficial to many fish species, so the relationship is complicated and should not be reduced to one of purely threat, but of impact more generally: Unravelling the macro-evolutionary ecology of fish-jellyfish associations. Caution is especially advised for poorly-understood freshwater jellyfish like *Limnocoeloides tangerinae*. Resource: life in the 'gingerbread house' Donal C. Griffin , Chris Harrod , Jonathan D. R. Houghton and Isabella Capellini
Published:20 March 2019<https://doi.org/10.1098/rspb.2018.2325>

23: What is your evidence that *Limnocoeloides tangerinae* have a noticeable sting to fish?

36-46: Consider the following sources and re-frame more accurately:

Flawed citation practices facilitate the unsubstantiated perception of a global trend toward increased jellyfish blooms Marina Sanz-Martín Kylie A. Pitt Robert H. Condon Cathy H. Lucas Charles Novaes de Santana Carlos M. Duarte First published: 22 June 2016
<https://doi.org/10.1111/geb.12474>

Questioning the Rise of Gelatinous Zooplankton in the World's Oceans Robert H. Condon, William M. Graham, Carlos M. Duarte, Kylie A. Pitt, Cathy H. Lucas, Steven H.D. Haddock, Kelly R. Sutherland, Kelly L. Robinson, Michael N Dawson, Mary Beth Decker, Claudia E. Mills, Jennifer E. Purcell, Alenka Malej, Hermes Mianzan, Shin-ichi Uye, Stefan Gelcich and Laurence P. Madin *BioScience* Vol. 62, No. 2 (February 2012), pp. 160-169 (10 pages)

64: There are thousands of Hydrozoan in the marine environment, many more than there are scyphozoans.

72: Comparing scyphozoans to freshwater hydrozoans is not really accurate, especially when hydrozoans are so abundant in marine systems. A more accurate comparison would be across hydrozoans.

74: The vast majority of hydrozoan medusae do not sting people and likely do not sting fish. What evidence do you have that fish can even perceive the sting of this poorly-know species?

77: But they may also be eaten. See: A Paradigm Shift in the Trophic Importance of Jellyfish? Author links open overlay panel Graeme C. Hays1 Thomas K. Doyle23 Jonathan D. R. Houghton4 Show more <https://doi.org/10.1016/j.tree.2018.09.001>

78: You can't make generalizations like this about jellyfish. As a clade Medusozoa has had more time to evolve and diversify than all of vertebrates combined.

83: The word invaded is inappropriate in this context. They are native and this is part of their natural cycle.

242: Wow, that's a lot of jellies!

247: Was the "sting" inferred from a behavioral response?

290: Stressor--a more general word would be perturbation.

380-382: Jellyfish may also serve as nurseries for young fish species: Ecological and Societal Benefits of Jellyfish Authors Authors and affiliations Thomas K. Doyle Graeme C. Hays Chris Harrod Jonathan D. R. Houghton Email author 1. 2. 3. 4. 5. 6. Chapter First Online: 28 September 2013

Citation 3 is for ctenophores, which are not jellyfish and is not related to the work being done here.

Author's Response to Decision Letter for (RSOS-191053.R0)

See Appendix A.

Decision letter (RSOS-191053.R1)

09-Oct-2019

Dear Dr Sefc,

I am pleased to inform you that your manuscript entitled "Freshwater hydrozoan blooms alter activity and behaviour of territorial cichlids in Lake Tanganyika" is now accepted for publication in Royal Society Open Science.

Before we do proceed to the production stage, we request that you please amend your competing interests statement; we have recently improved our policies for transparency, and request that manuscripts which list an Associate or Subject Editor as a co-author declare this within the competing interests statement. Please therefore update this within your manuscript as follows:

"Professor Kristina Sefc is a Board Member of Royal Society Open Science"

We would be grateful if you could then send us the updated copy of your manuscript document as an attachment in reply to this email -- thank you for your help with this.

on behalf of Dr Punidan Jeyasingh (Associate Editor) and Professor Kevin Padian (Subject Editor)
openscience@royalsociety.org

Associate Editor Comments to Author (Dr Punidan Jeyasingh):

I thank the authors for thoroughly addressing reviewer comments. This version is much improved, and ready for press. Congratulations to the authors. This is a fascinating line of study that is developing. I'll be following it closely. All the best for future efforts.

Follow Royal Society Publishing on Twitter: [@RSocPublishing](https://twitter.com/RSocPublishing)
Follow Royal Society Publishing on Facebook:
<https://www.facebook.com/RoyalSocietyPublishing.FanPage/>
Read Royal Society Publishing's blog: <https://blogs.royalsociety.org/publishing/>

Appendix A

Institute of Biology

Universitätsplatz 2, 8010 Graz
Phone: +43 316 380 – 5601
Email: kristina.sefc@uni-graz.at
<https://homepage.uni-graz.at/de/kristina.sefc/>

Editor-In-Chief

Dr. Spencer Barrett

September 9, 2019

Dear Dr. Barrett,

We are grateful to you and the editors (Dr. Punidan Jeyasingh and Kevin Padian) for facilitating the review of our manuscript (RSOS-191053). We also thank the anonymous reviewers for their positive and insightful comments. We have revised the manuscript according to their suggestions. In brief, we have considerably rewritten the introduction and portions of the discussion, to better represent the field's current view on the threats and/or benefits of jellyfish blooms. We have also included more information on how we differentiated between the 'bloom' and 'non-bloom' conditions, as well as how we assessed the stinging potential of our focal jellyfish species, *Limnocnida tanganicae*. Below, we have responded to each of the reviewer's comments, leaving their comments in plain text and writing our responses in bold text. Furthermore, within our manuscript files, the changes to our text have been highlighted.

We thank you again for your assistance and look forward to hearing from you at your earliest convenience.

Sincerely,
Kristina M Sefc (corresponding author)

Reviewer: 1

This is a good, well written paper dealing with the behavioural impacts of a freshwater hydrozoan on a territorial cichlid and on the composition of cichlids assemblages. The main issue with the manuscript is that authors have chosen a highly debatable or wrong context to justify their study, which in my view, they don't need: the study is interesting and valid without establishing a parallel with fish-jellyfish interactions in coastal marine systems. Despite the diversity of studies suggesting that anthropogenic stressors facilitate jellyfish blooms and a global trend of increasing jellyfish blooms, two recent studies (Pitt et al. 2018; Sanz-Martín et al. 2016) have clearly shown that the evidence supporting those claims is weak and mainly based in a couple of cosmopolitan species. These species do not represent the huge diversity life forms included in the term “jellyfish”, which depending on the authors may comprise between two (Cnidarians, Ctenophores) and five phyla (Cnidarians, Ctenophores, Molluscs, Polychaetes and Chordata-Salps and Larvaceans). Therefore, I strongly recommend not to use this parallel to justify the study or to discuss the results.

> Thank you for the positive thoughts and constructive comments on our manuscript. We hope that we have addressed all of your concerns below.

Specific comments

Title: It is too general; it does not reflect the scope of the study. The manuscript deals with the effect of a freshwater hydrozoan on the behaviour of cichlids in the Tanganyika Lake.

> The title has been changed to better reflect the scope of our study: Freshwater hydrozoan blooms alter activity and behaviour of territorial cichlids in Lake Tanganyika

Introduction:

- I suggest either avoid the use of the term “jellyfish” or clearly define it in the context of this study

> We now use the term “gelatinous zooplankton” when discussing the literature in general, and we use the term “jellyfish” when specifically referring to *L. tanganyicae*.

- I suggest, if available, to include information on the abundance seasonality of the *L. tanganyicae*

> Unfortunately, there are no data pertaining to the seasonality of *L. tanganyicae* blooms. Overall, this species has received very little research attention. We now emphasize this point in the text.

L34-63. See general comments and reconsider the need of this section.

> Thank you, we have extensively rewritten our introduction and in-so-doing changed our justification and motivation for our study. This version is a more accurate representation of the current state of the field.

L88. Please define -here or in material and methods- how a bloom and non-bloom condition was defined. It was not clear for me if “normal” or “non-bloom” conditions mean no medusas at all or a low density.

> We have now added more information on how we differentiated the two conditions. Lines 74-76.

Results:

L243-244. This is an interesting result, suggesting that the medusae population was mainly comprised of juveniles –this, by the way was not discussed in the next section. I am wondering if there is differences between juvenile and adult medusa in prey patterns and interspecific competitive interactions.

> We have avoided attempting to classify *L. tanganjicae* into different life stages (e.g. juvenile vs. adult). However, we raise the interesting point that *L. tanganjicae* were apparently ignored as a food source despite being of the appropriate prey size for our study fish. Lines 282-283. It would certainly be interesting to look at how fish-jellyfish interactions might differ across size classes of both species, but we were unable to gather these data here for this question.

Discussion:

Generally I suggest avoiding stating again the results in order to discuss about them. The discussion is a bit too long.

> We have reduced the overall length of the discussion, while keeping all relevant discussion points.

L 274-280: Please see my general comments.

L280. In my view this section should start here

> We have removed this first portion of the discussion as it was no longer relevant to our study's justification and motivation. As suggested, we start the discussion at the indicated line.

L 307-310: This paragraph should be at the end. Of this section

> We have now moved this section of the text to the end of the discussion as suggested.

L373-380: This sections sounds to me as “concluding remarks”, please consider

> It is not immediately clear what the reviewer means here. However, we have rewritten portions of this paragraph in the general spirit of the rest of their comments.

L381-384: see general comments

> This section of text has now been removed.

Figures:

Figure 2. Please revise the term “affiative” in the X-label

> Thank you for catching this typo. It has been corrected.

References:

Pitt, K. A., Lucas, C. H., Condon, R. H., Duarte, C. M., & Stewart-Koster, B. (2018). Claims that anthropogenic stressors facilitate jellyfish blooms have been amplified beyond the available evidence: a systematic review. *Frontiers in Marine Science*, 5, 451.

Sanz- Martín, M., Pitt, K. A., Condon, R. H., Lucas, C. H., Novaes de Santana, C., & Duarte, C. M. (2016). Flawed citation practices facilitate the unsubstantiated perception of a global trend toward increased jellyfish blooms. *Global Ecology and Biogeography*, 25(9), 1039-1049.

> We have now included these very useful references.

Reviewer: 2

Comments to the Author(s)

This is a fascinating study on the impact of a poorly-known jellyfish species on fish behavior. Overall the manuscript is well-written, but reflects several outdated views on jellyfish blooms which, even if accurate, are not comparable to this system. Unlike highly impacted coastal marine areas, the author's study area is an ancient freshwater lake in which the jelly species in question is a native member of the ecosystem. While it is true that jellyfish blooms impact fish behavior in marine systems, framing this study in the context of 'rising jellyfish numbers' and the 'global rise of jellyfish' (claims that are now being questioned), detracts from the fascinating work being done here on an remarkable and unique system. Because the human-associated causes and consequences of marine blooms likely do not impact this system, and because the field of marine blooms is itself rapidly changing, the authors should re-frame their introduction and discussion to highlight their system's fascinating qualities. The fact that so little is known about freshwater jellyfish ecology, and this species in particular, should be celebrated. Further, I see no reason their findings are not applicable to marine systems, but the language and literature currently cited reflect a "jellyfish as negative" perspective when really these are all native species interacting in a fascinating way.

> Thank you for the encouraging review. We have addressed the points that you raise below and think that the changes have made significant improvements to the manuscript.

15: Jellyfish are also beneficial to many fish species, so the relationship is complicated and should not be reduced to one of purely threat, but of impact more generally: Unravelling the macro-evolutionary ecology of fish–jellyfish associations. Caution is especially advised for poorly-understood freshwater jellyfish like *Limnocnida tanganjicae*. Resource: life in the 'gingerbread house' Donal C. Griffin , Chris Harrod , Jonathan D. R. Houghton and Isabella Capellini Published:20 March 2019<https://doi.org/10.1098/rspb.2018.2325>

> We have rewritten a great deal of our introduction and discussion to downplay the threats of jellyfish blooms and better convey the current state of the field.

23: What is your evidence that *Limnocnida tanganjicae* have a noticeable sting to fish?

> Our evidence is behavioural. We noticed the fish making sudden, quivering or jolting movements during the jellyfish bloom conditions, which we did not observe during the non-bloom conditions. We have now emphasized this in the text (lines 68, 268) and in Table 1. The supplementary video shows an example of a fish jolting in reaction to contact with a jellyfish.

36-46: Consider the following sources and re-frame more accurately:

Flawed citation practices facilitate the unsubstantiated perception of a global trend toward increased jellyfish blooms Marina Sanz- Martín Kylie A. Pitt Robert H. Condon Cathy H. Lucas Charles Novaes de Santana Carlos M. Duarte First published: 22 June 2016 <https://doi.org/10.1111/geb.12474>

Questioning the Rise of Gelatinous Zooplankton in the World's Oceans Robert H. Condon, William M. Graham, Carlos M. Duarte, Kylie A. Pitt, Cathy H. Lucas, Steven H.D. Haddock, Kelly R. Sutherland, Kelly L. Robinson, Michael N Dawson, Mary Beth Decker, Claudia E. Mills, Jennifer E. Purcell, Alenka Malej, Hermes Mianzan, Shin-ichi Uye, Stefan Gelcich and Laurence P. Madin *BioScience* Vol. 62, No. 2 (February 2012), pp. 160-169 (10 pages)

> Thank you, we have now revised the manuscript according to these sources (see also references #1, 3, 5).

64: There are thousands of Hydrozoan in the marine environment, many more than there are scyphozoans.

> **We have now removed this sentence from the manuscript.**

72: Comparing scyphozoans to freshwater hydrozoans is not really accurate, especially when hydrozoans are so abundant in marine systems. A more accurate comparison would be across hydrozoans.

> **This portion of the text has also been removed.**

74: The vast majority of hydrozoan medusae do not sting people and likely do not sting fish. What evidence do you have that fish can even perceive the sting of this poorly-known species?

> **Please see our above comment (for line 23).**

77: But they may also be eaten. See: A Paradigm Shift in the Trophic Importance of Jellyfish? Author links open overlay panel Graeme C. Hays¹ Thomas K. Doyle^{2,3} Jonathan D. R. Houghton⁴ Show more <https://doi.org/10.1016/j.tree.2018.09.001>

> **Thank you, we have now included this important reference into our manuscript. Line 40, reference #13.**

78: You can't make generalizations like this about jellyfish. As a clade Medusozoa has had more time to evolve and diversify than all of vertebrates combined.

> **It is not clear what the reviewer is referring to here. However, we have changed our wording here to emphasize that we are talking about *L. tanganyicae*. Lines 65-66.**

83: The word invaded is inappropriate in this context. They are native and this is part of their natural cycle.

> **We have now changed the wording here.**

242: Wow, that's a lot of jellies!

247: Was the "sting" inferred from a behavioral response?

> **Yes, we inferred the sting from the behavioural responses of the fish. This is now made clearer in the text (lines 68, 268) and in Table 1.**

290: Stressor--a more general word would be perturbation.

> **We have changed "stressor" to "perturbation" in the text.**

380-382: Jellyfish may also serve as nurseries for young fish species: Ecological and Societal Benefits of Jellyfish Authors Authors and affiliations Thomas K. Doyle Graeme C. Hays Chris Harrod Jonathan D. R. Houghton Email author 1. 2. 3. 4. 5. 6. Chapter First Online: 28 September 2013

> **Thank you, we have now also included this reference in our manuscript.**

Citation 3 is for ctenophores, which are not jellyfish and is not related to the work being done here.

> **Thank you, this reference has been removed.**